# A Needs Learning Algorithm Applied to Stable Gait Generation of Quadruped Robot

**DOI:** 10.3390/s22197302

**Published:** 2022-09-26

**Authors:** Hanzhong Zhang, Jibin Yin, Haoyang Wang

**Affiliations:** Faculty of Information Engineering and Automation, Kunming University of Science and Technology, Kunming 650221, China

**Keywords:** machine learning, Maslow’s hierarchy of needs, demand decision

## Abstract

Based on Maslow’s hierarchy of needs theory, we have proposed a novel machine learning algorithm that combines factors of the environment and its own needs to make decisions for different states of an agent. This means it can be applied to the gait generation of a quadruped robot, which needs to make demand decisions. To evaluate the design, we created an experimental task in order to compare the needs learning algorithm with a reinforcement learning algorithm, which was also derived from psychological motivation theory. It was found that the needs learning algorithm outperformed the reinforcement learning in tasks that involved making decisions between different levels of needs. Finally, we applied the needs learning algorithm to the problem of stable gait generation of quadruped robot, and it had achieved good results in simulation and real robot.

## 1. Introduction

Since the beginning of artificial intelligence as an academic discipline in the 1950s, it has been continuously developing and has become an increasingly important part of the technology industry. In particular, reinforcement learning has become one of the three main machine-learning paradigms, along with supervised learning and unsupervised learning [1]. Reinforcement learning focuses on how an agent should act in an environment to maximize cumulative rewards. It is suitable for solving problems that involve weighing long-term and short-term rewards. Reinforcement learning is currently used in many fields, such as computerized play in the game of Go [2], autonomous driving [3], and quadrupedal robots [4].

Quadrupedal bionic robots are those that imitate the shape and gait of quadrupeds in nature to accomplish specific actions or tasks. However, at present, manually designing controllers for quadruped robots that perform similarly to animals is a very difficult task. Designing controllers manually requires constructing a large number of formulae and performing complex matrix transformations, as well as extensive derivation of formulae and tedious manual adjustments during the design process. When implemented on a physical robot, these methods also need to address random noise and data transmission delays due to hardware problems [5].

In recent years, data-driven approaches, such as Deep Q-Network (DQN), have proven to be promising approaches to overcome the limitations of previous model-based approaches, and to develop effective motor skills for robots. Lee et al. [6] trained a controller for legged locomotion on challenging terrain via RL in simulation and showed its robustness under real-world conditions, which has never been encountered in simulation training. Sun et al. [5] proposed a convenient and adaptable method to construct a self-balancing controller for a quadruped robot. The method used RL and ANN for policy design, eschewing the construction of kinematic equations, simplifying the design process and enhancing the adaptability of the control strategy.

In fact, when stable gait generation gradually becomes the key point in the research into quadruped robots, it first exists as a demand decision-making problem that needs to balance the two-level demands (maintaining stability or movement) [7]. However, reinforcement learning, as one of the cores of DQN algorithms, has some inadaptability in this problem.

Reinforcement learning is mainly derived from the behaviorist reinforcement theory of motivation in psychology. Motivation refers to the behavior of organisms that is affected by three factors: (1) an irresistible external influence; (2) an internal demand, motivation, plan, etc.; and/or (3) an external object or situation that serves as a goal or motivation. The first factor is largely independent of the internal state of the organism and is called extrinsic motivation; the latter two factors involve hypothesized internal states that are thought to be necessary to explain behavior and are called intrinsic motivation.

Baldassarre [8] explores motivation from a biological perspective. He argues that extrinsic motivation refers to doing something because of some externally provided reward, while intrinsic motivation refers to “doing something because it is interesting or pleasant”. From this perspective, reinforcement motivation theory focuses on extrinsic motivation and discards an individual’s other considerations, that is, it ignores intrinsic motivation.

In psychology, from a non-behaviorist perspective, more attention is paid to the study of people’s normal behavior within a larger context, and the influence of human needs on motivation has become a key research direction. The hierarchy of needs theory was proposed to explain the differences in the influence of different human needs on motivation. The theory comes from Humanism, one of 4 schools of psychology. Behaviorism, the theoretical source of reinforcement learning, also belongs to one of them. The motivation theories corresponding to the four schools of psychology and their corresponding algorithms are shown in Figure 1.

Under such a premise, reinforcement learning can show some adaptability to the problem of stable gait generation of quadrupedal robots, and then needs learning, a method with similar origins to reinforcement learning and specializing demand decisions, should be able to show better results than reinforcement learning.

In this regard, it is particularly necessary to use this existing demand motivation theory, which is parallel to the reinforcement motivation theory, to propose a needs learning algorithm that can be applied to process data with a hierarchy of demand priorities in different environments. Additionally, in this paper, needs learning will be applied to solve the stable gait generation problem of a quadruped robot to verify its effectiveness compared with reinforcement learning in this problem.

### 1.1. Related Work

It is generally believed that reinforcement learning can only deal with extrinsic motivation, but there are still some methods that combine reinforcement learning with intrinsic motivation. Barto [9] pointed out that possibility and the importance of introducing intrinsic motivation into a reinforcement learning framework. Kulkarni [10] proposed an RL framework that combines a hierarchical action value function that operates at different time scales with goal-driven deep reinforcement learning.

Although researchers have made some efforts in this area, the main mechanism of reinforcement motivation theory is adaptation to different combinations of external stimuli to produce appropriate reflexive responses [11]. This work determined that reinforcement learning could not demonstrate its superiority in tasks in which it had to weigh its own needs against the needs of a target. Such tasks are particularly significant in the computer field, especially in agent control and other issues.

### 1.2. Contributions

This paper mainly completed the following work:A needs learning algorithm is proposed based on the hierarchy of needs theory, which makes it possible to present good results for problems that require weighing different levels of needs;The proposed needs learning algorithm is compared with traditional reinforcement learning, in order to obtain a better task completion effect, so as to verify its effectiveness;The needs learning algorithm is applied to the gait generation of quadruped robot, and good application results are obtained in simulation and real robot.

In Section 2 of this paper, we will introduce, in detail, Maslow’s hierarchy of needs on which the needs learning algorithm relies, and put forward the algorithm model and process of needs learning; Section 3 will compare the needs learning algorithm and reinforcement learning algorithm in the classic task of cart pole to verify their effectiveness; and in Section 4, the needs learning algorithm is applied to the gait generation task of quadruped robot, and good results are achieved in simulation and real robot testing.

## 2. Method

In this section, we first introduce Maslow’s hierarchy of needs as the basis of the needs learning algorithm in Section 2.1, and then put forward the specific description of the needs learning algorithm in Section 2.2.

### 2.1. Maslow’s Hierarchy of Needs

#### 2.1.1. Hierarchy of Needs

Maslow [12] originally proposed the hierarchy of needs theory in 1943. He suggested that need is the source of motivation and that the intensity of the need determined the intensity of the motivation.

In his theory, Maslow indicated that people have two types of needs: basic needs and growth needs. Basic needs are driven by human instincts and sustain human survival; growth needs represent human sociality, including dignity, love, self-realization, and so on. In the hierarchy of needs theory, human needs are diverse, but only the most dominant needs become the main motivation for behavior, and only after lower-level needs are satisfied will people pursue higher-level needs.

The needs levels are considered to develop in waves; they do not develop as a step function but rather continually increase until satisfied. Different levels of various needs exist at the same time, and higher-level needs do not necessarily appear after lower-level needs reach a maximum. The relationship between the development of each level of needs is shown in Figure 2.

At different periods, different levels of needs have different influences on human motivation. The need that has the greatest power to control behavior is called the dominant need. A change in the hierarchy of needs in Maslow’s theory is considered a “change in the dominance of needs” rather than a “change in the needs” [13].

#### 2.1.2. Self-Realization

Self-realization is indicated by individuals striving to be what they can be. It is at the top of the hierarchy of needs and is the weakest need, which is easily suppressed by lower-level needs. Self-realization is achieved by enjoying life and devoting oneself to a career wholeheartedly.

Peak experience is a kind of blissful instantaneous experience when you enter a state of self-realization and self-transcendence. There are two types of peak experiences: the ordinary type and the self-realization type. The ordinary type of peak experience is when self-realization is not yet finished, which can also be called a partial peak experience, while the self-realization type of peak experience is a composite state of an identity experience and cognition of being [14].

When a person reaches a peak experience, it can weaken the inhibition of low-level needs on high-level needs and let individuals spend more resources on high-level needs.

#### 2.1.3. Motivation and Behavior

Maslow believed that needs are the source of motivation, and the intensity of the needs determines the intensity of the motivation. Motivation results in coping behaviors, and Maslow believed that coping is the result of acquired learning. The decisive impulse of coping behavior includes needs, goals, intentions, functions, and purposes. Most behaviors are driven by multiple motives.

There are two types of human behavior: coping behaviors and expressive behaviors [15]. Coping behaviors are purposeful and motivated, and expressive behaviors are not. Most coping behaviors are the result of acquired learning and require rewards to maintain, while expressive behaviors are innate. We can understand expressive behaviors as mechanical reflexes, such as finger lifting, and coping behaviors as behaviors in a broader sense, which include running, eating, etc.

Bobick [16] stated that human movement can be divided into movement, activity, and action, while Liguo [17] suggested that human movement can be divided into behavior, action, and basic action. They both described action as the basic unit of human movement and that action constitutes more complex actions and behaviors. For example, a finger-bending action is part of grabbing; grabbing along with other actions constitute eating behavior.

The characteristics of expressive behavior are consistent with the characteristics of basic actions, while the characteristics of responsive behavior are consistent with the characteristics of the behavior in human movement. Therefore, we believe that in the acquired learning of coping, there are different combinations of expressive behaviors in different environments. Through rewards from the environment, a person can learn coping behaviors to adjust to the environment.

Based on the above discussion, we propose a needs learning algorithm based on the hierarchy of needs.

### 2.2. The Needs Learning Algorithm

We designed needs learning as an environment exploration algorithm based on dynamic programming. We set up an agent and assume that it can think independently and learn to make behavior decisions according to its own task needs and its external environment state as it continuously explores the environment.

In dynamic programming, the state at the current stage is often the result of the state and decision at the previous stage. If the state $S_{k}$ at stage *k* and the decision $u_{k}(S_{k})$ are given, then the state $S_{k+1}$ at stage *k* + 1 is also completely determined. That is, there is an explicit quantitative correspondence between $S_{k+1}$ and both $S_{k}$ and $u_{k}$. We denote this as $T_{k}(S_{k},\;u_{k})$, that is:(1)$$S_{k+1}=T_{k}(S_{k},\;u_{k})$$

This equation, which represents the relationship between a state and its previous state as a function, is called the state transition equation.

We first need to explain the vocabulary of Maslow’s hierarchy of needs, as it is applied in our algorithm. The terms and their explanations are shown in Table 1.

According to Maslow’s hierarchy of needs and the process of human responses in exploring an environment, we could provide the algorithmic process of needs learning. A schematic diagram of the algorithmic process is shown in Figure 3.

#### 2.2.1. The Wave Model

Lower-level needs can inhibit higher-level needs. Maslow referred to the needs development relation in Figure 2 as the wave model. However, since Maslow and subsequent psychologists have not given a specific function for the wave model, we define it as follows:

For the function used to represent each level, the independent and dependent variables are independent of those in the other levels. The specific independent variable values are set according to the problem, and the independent variables required by the nth level are normalized into [0, 1 + 0.5(n − 1)]. The dependent variable represents the value of the demand and is normalized into [0, 1]. The larger the value of the dependent variable, the more the agent will need to meet this hierarchy of needs.

Since each need $F_{n}$ is inhibited by a need at a lower level, we considered that the original functional form $f_{n}$ is consistent for each level of need. Additionally, for each level of need, $f_{n}\left(0\right)=1$, and, subsequently, the need value decreases continuously until zero. That is, the agent will gradually decrease its desire for a need as it is satisfied. For the level of needs with *n* > 1, there is an inhibition relationship between the lower-level function values on it. For details, see Formula (2).
(2)$$F_{n}\left(x\right)=f_{n}\left(x\right)-\overset{n-1}{\underset{m=0}{\sum }}f_{m}\left(x\right)$$

The function $f_{n}$ can be represented by a logistic function, i.e.:(3)$$f_{n}\left(x\right)=\frac{1}{1+e^{10cx-5n}}$$

The value c of this function represents self-realization when used to calculate the inhibition of higher-level needs. A larger c represents a larger value of self-realization, and the function is compressed on the independent variable scale to reduce $f_{n}\left(x\right)$ and, in turn, reduces the inhibition to the high-level needs. Let $c=1+\alpha x_{\mathrm{GNS}}$. The variable $x_{\mathrm{GNS}}$ is a value of the independent variable of growth needs, and α is the base of the self-realization inhibition. The raw wave function images of *n* = 1 and c = 1 are shown in Figure 4a. The graph of the inhibition relationship for the different needs levels is shown in Figure 4b.

After the growth needs value and the basic needs value are calculated by the wave function, the higher one is defined as the dominant need.

#### 2.2.2. Coping Behavior

Herbert Simon put forward the “theory of bounded rationality”. He believed that classical decision theory, which is the reference theory of most machine learning, always tries to build an “absolutely rational” model, so that agents can constantly approach the only optimal solution in the environment. However, in the real decision-making of human beings, rationality is limited, and it is not worth the loss of wasted resources and computing power to pursue the unreachable “absolute optimal solution”. Decision models, especially those containing demand objectives, should be based on the premise of human bounded rationality to obtain the most cost-effective [18].

Therefore, in the algorithm optimization process of needs learning, this paper referred to the learning process of human behavior in Maslow’s hierarchy of needs, and tried to make it algorithmic.

We have previously noted that, in the acquired learning of coping, there are different combinations of expressive behaviors in response to environmental changes. Through environmental rewards, agents learn coping behaviors. In this regard, we can specify the existence of n different expressive behaviors $a_{n}$ and define a coping behavior that contains only one expressive behavior stored in a set of executable behaviors $A=\left\{A_{n},\;n\in N^{*}\right\}$, where $A_{n}=\left\{a_{n}\right\}$. The variable $A_{n}$, as a coping behavior, represents an ordered set of actions that the agent may perform in state $S_{i}$.

We first set a maximum exploration probability $r_{max}$. Let $r_{max}=1$, and let it decrease with the number of iterations. Before the start of each response to the environmental state $S_{i}$, we randomly select two coping behaviors $A_{n}$ and $A_{m}$. Because more complex behavior is harder to learn, we set a decay function, and set the exploration probability as $r=\frac{r_{max}}{2l-1}$, where l is the sum of the lengths of $A_{n}$ and $A_{m}$. We generate a random number in the range [0, 1), and if it is smaller than r, $A_{n}$, and $A_{m}$ are aggregated to produce a new coping behavior $A_{k}=A_{n}+A_{m}$, which is stored in A and executed.

After executing the behavior, we calculate the amount of improvement in the environment, in other words, the degree of state migration to the target state after the execution of $A_{k}$, which is called M. We define an M-table for each level of needs. After executing $A_{k}$ at state $S_{i}$, the M-value is stored in the M-table of the dominant need.

When the generated random number is larger than r, we select a behavior with the largest M-value in the current state in the M-table and execute it. If the data of that state do not exist in the M-table, we let $A_{k}=A_{n}$ and execute it.

In conclusion, we can obtain Algorithm 1 flow diagram of the needs learning.
**Algorithm 1** Needs learning algorithm Initialize the independent variable of basic needs xBNS and the one of growth needs xGNS Initialize the maximum exploration probability rmax Initialize *M* tables of basic needs and growth needs Initialize *c* = 1** Repeat** (for each episode):  r ← rmax2l−1  xBNS ← 11+e10cfBNSs  xGNS ← 11+e10cfGNSs−5−xBNS*  c* ← 1+ɑxGNS**  If** xGNS>xBNS **then**    Dominant Need ← growth needs**  Else**    Dominant Need ← basic needs  Choose Am,An from *A* at random  Choose r0 randomly from the interval [0, 1)**  If**
r0 < r **then**    Ak = {Am,An}    Save Ak into *A***  Else****    If**
*s* in *M* table of dominant need **then**      Take Ak with the largest *M* value from the *M* table of dominant need**    Else**     Ak ← An  Execute action Ak to obtain the environmental improvement amount Ms  Update *M* table of dominant need with Ak, *s*, Ms


## 3. Algorithm Comparison

In recent years, reinforcement learning has been widely used in various fields and has achieved impressive results due to its powerful exploratory and autonomous learning capabilities. At the core of reinforcement learning is the study of the interaction between an intelligent body and its environment in order to make sequential decisions that will obtain a maximum reward by continuously learning an optimal strategy [19].

Watkins [20] proposed the Q-learning algorithm in 1992, which is one of the most main algorithms in reinforcement learning. To explore an environment, its core is using a formula to update a Q-table and selecting the action with the highest utility based on the table. The formula for updating the Q-table is:(4)$$\mathrm{NewQ}\left(s,a\right)=Q\left(s,a\right)+\alpha \left[R\left(s,a\right)+\gamma {\mathrm{maxQ}}^{\prime }\left(s^{\prime },a^{\prime }\right)-Q\left(s,a\right)\right]$$

α is the learning rate, γ is the discount factor, and ${\mathrm{maxQ}}^{\prime }\left(s^{\prime },a^{\prime }\right)$ represents the maximum utility that the agent can obtain in the $s^{\prime }$ state.

Reinforcement motivation theory, which underlies reinforcement learning, and Maslow’s hierarchy of needs theory are two of the major motivational reinforcement theories in psychology [13]. It was necessary to verify the effectiveness of needs learning by comparing them. We designed two tasks to compare their actual effects.

Through the Python simulation library Gym, we designed a task that needs learning and Q-learning could each complete, so that we would be able to compare the results.

Cart pole is a game in which the goal is to keep a pole upright (and not topple over) by pushing the body on which it stands left or right.

We defined the task as follows: let the angle of the pole deviate not more than 0.2 units and let the body move to the right side of the interface 0.5 units from the origin. The description of this task is shown in Figure 5.

We let reinforcement learning (RL) and needs learning (NL) iterate 5000 times each, and executed up to 1000 actions per iteration.

We wrote a program using the Q-learning algorithm and set its learning rate to 0.8 with a discount factor of 0.8 and a random action probability of 0.9 (it would decrease linearly to zero as it iterated). We also defined its environmental rewards: −0.1 for each additional action order executed, d × 300 for moving distance d, and −200 for falling.

At the same time, we also wrote a needs learning program and set the self-realization inhibition base to 0.8, the independent variable of basic needs to $x_{\mathrm{BNS}}=5\times \left|a\right|$, and the independent variable of growth needs to $x_{\mathrm{GNS}}=2\times \left|l\right|$. The variable a was the rotation angle of the pole in the current state, and l was the distance from the body to the origin.

After completing 5000 iterations each, the success of reinforcement learning versus needs learning and their required number of steps were compared, as shown in Figure 6a. The comparison of the steps of a single iteration is shown in Figure 6b.

In Figure 6a, each peak represents a successful completion of the task, and the lower the peak, the lower the number of actions used to complete the task.

It is clear from Figure 6a that the completions in the needs learning approach were more intensive than in reinforcement learning in all the iteration periods, and needs learning clearly appeared to be successful earlier than the reinforcement learning, which means that needs learning had a faster learning rate (in terms of iterations) and a better learning effect than reinforcement learning in this problem.

In Figure 6b, by and large, the broken line of needs learning was higher than that of reinforcement learning, which means that needs learning had learned general strategies that could move the object farther than reinforcement learning in this problem.

The comparison experiments were about weighing the agent’s own needs against the target’s needs. Needs learning showed better results than reinforcement learning when faced with intrinsic needs, such as maintaining the agent’s survival.

By comparing the results, we can conclude that needs learning showed better results than reinforcement learning in problems that required weighing the agent’s own needs against the target’s needs.

## 4. Experiment

After verifying that the needs learning algorithm can show its effectiveness in the tasks that need to make demand level decisions, we hope to apply it to the quadruped robot stable gait generation task. In this section, we will first deploy it to the simulation environment for testing, and also use reinforcement learning for the same task. After the comparison has achieved relatively good results, we deploy the trained needs learning model to the quadruped robot real robot for testing to ensure that it has the same excellent effect in the simulation environment and real robot. The simulation and real machine are shown in Figure 7.

### 4.1. Simulation

PyBullet is a quadruped robot simulation library for Python that can manipulate the 12 motors of a virtual quadruped robot to perform many tasks within a simulated environment. This includes gait simulation, torso stabilization, etc.

With PyBullet, we defined a complex task: let the quadruped robot move as fast as possible to a position at a distance of 1 unit from the origin without falling; in other words, with the *z*-axis coordinate of the center of gravity not less than 0.1 units, let reinforcement learning and needs learning iterate 200 times each, executing up to 1000 actions per iteration.

In reinforcement learning, although model free learning performs poorly in the initial state, it can often learn the optimal solution outside the rules [21]. We expect needs learning to have a similar effect, so initially we set the state space as an empty set, and the random strategy is that the 12 motors of the quadruped robot rotate forward or backward by an angle that can be 0, so as to make it learn gait.

We wrote a program using the Q-learning algorithm and set its learning rate to 0.8 with a discount factor of 0.8 and a random action probability of 0.9 (it would decrease linearly to zero as it iterated). We also defined its environmental rewards: −0.1 for each additional action order executed, d × 100 for moving distance d, and −200 for falling.

At the same time, we wrote a needs learning program and set the self-realization inhibition base to 0.8, the independent variable of basic needs to $x_{BNS}=\frac{5-10z}{4}$, and the independent variable of growth needs to $x_{GNS}=l$. The variable z was the *z*-axis coordinate of the center of gravity of the quadruped robot, and l was the distance from the body to the origin.

After completing 200 iterations each, the success of reinforcement learning versus needs learning and their required number of steps were compared, as shown in Figure 8a. The comparison of the steps of a single iteration is shown in Figure 8b. The failure (fall) occurrences of the quadruped robot are shown in Figure 8c.

In Figure 8a, each peak represents a successful completion of the task, and the lower the peak, the lower the number of actions used to complete the task.

It is clear from Figure 8a that the completions of needs learning were more intensive than in reinforcement learning in all the iteration periods, and needs learning clearly appeared to be successful earlier than the reinforcement learning, which means that needs learning had a faster learning rate (in terms of iterations) and better learning effect than reinforcement learning in this problem.

Figure 8b represents the final distance moved by the agent at the completion of each iteration, and its value of 1 represents the completion of the task. Although the magnitude of change in the oscillation of needs learning was larger than in that of reinforcement learning, as seen in Figure 8b, the optimal case search and average final number of steps of needs learning were better than those of reinforcement learning, which means that needs learning learned a strategy that could move farther than reinforcement learning in this problem. It can be inferred from Figure 8c that the shorter moving distance of needs learning may be a strategic trade-off to satisfy the need for the quadruped robot to remain standing.

What is represented in Figure 8c is the case of task failure (the agent body falls), not the case of task non-completion (not walking to the end). It can be seen that reinforcement learning had frequent falls during the exploration of the environment, while needs learning never appeared to be unable to maintain standing. We could assume that needs learning was able to make a more favorable trade-off for itself when facing different demand conflicts. That is, it is more effective in weighing the needs of survival (not falling) and mission (moving forward).

From this, it can be concluded that need learning accomplished this task better than reinforcement learning.

### 4.2. Real Robot

There is a certain difference between the simulation environment and the real world. Without adaptive debugging, the training results of the simulation environment can only run well in the simulation environment, and cannot be moved in the real world. In order to make the robot motion in the simulation environment almost consistent with the robot motion in the real environment, this paper uses Raspberry Pi and IMU attitude sensor to make a quadruped robot, and tries to deploy the algorithm for testing.

The real robot used in this paper was designed in 3D using Unigraphics software to divide the whole machine into two parts: the torso controller and the series legs. The series legs part mainly consists of hip joint support, hip joint motor, knee joint, knee joint motor, motor pull rod, and foot [22].

The hip joint of the series leg uses a disc structure as a support to increase the stability of movement. Meanwhile, the hip joint motor is positioned through a pre-set hollow slot, fixed to the flange plate by M2 self-tapping screws, and connected to the whole thigh. The knee joint uses an F683 flange bearing and cylindrical pin to connect the thigh and crus. The crus drive uses the rudder horn pull rod structure, which makes the motor position move up and reduces the inertia of the leg movement. A rubber shock-absorbing groove is reserved at the foot end, which is convenient for pasting the rubber pad, and the structure is hollowed out to reduce the inertia of the leg movement.

The driver, 2S lithium battery, Raspberry Pi, power module, DC plug, and other components are placed in the torso of the quadruped robot. It uses a buck module for voltage reduction and uses a 2S lithium battery and 12VDC mixed power supply. The rotation angle of the motor is calculated by the motion pos function, and the underlying driver communicates with the PCA9685 using the I2C interface, so that it outputs a 50 HZ PWM signal to drive the motor. The final quadruped robot for testing is shown in Figure 7b.

In the real robot, we used the needs learning motion control model trained in the simulation environment in Section 4.1 to test the comparison of the model on the real robot and in the simulation. The control policy is shown in Figure 9.

In the real robot test, after carrying the gait generated by the needs learning algorithm, the motion trajectory of the quadruped robot is shown in Figure 10.

The *y*-axis represents the left–right offset of the quadruped robot. The maximum left–right offset is 25 mm, which indicates that the quadruped robot can roughly keep walking in a straight line; the *z*-axis represents that the quadruped robot can pitch up and down at a maximum of 40 mm. At this time, the center of mass of the body does not float up and down relative to the height of the ground, indicating that the quadruped robot moves smoothly. The negative direction of the *x*-axis represents the forward direction of the quadruped robot. It has walked for 8 motion cycles in 3 s, moving forward a total of 175 mm, and the walking speed is about 0.0583 m/s. The small error is due to the certain sliding with the ground during the walking process of the quadruped robot. The motion speed is close to the theoretical speed, and the curve is smooth, which shows that the quadruped robot can move forward harmoniously and smoothly. The gait of quadruped robot generated by needs learning algorithm is shown in Appendix A, which is supplied in Appendix A.

The above results show that the stable gait generated by needs learning can make the quadruped robot have a good effect consistent with the simulation environment.

### 4.3. Discussion

Needs learning utilizes the mechanism of human behavioral combinations and demand trade-offs, thus gaining the ability to trade-off actions for different needs. The experiment showed that needs learning improved over reinforcement learning in strategy selection and learning speed. This certainly provides an effective method for tasks that require demand decisions, such as unmanned driving and robot gait.

From the task of quadruped robot movement, we can see that needs learning could effectively and quickly change its own strategy to ensure safety in the face of high-risk situations. The current research on needs learning is still in the preliminary stage, and there is still a gap in effectiveness compared to the state-of-the-art development for gait generation in quadruped robots. However, as a new algorithmic model, needs learning presents advantages over Q-learning, a traditional reinforcement learning method, and some improvements for reinforcement learning (e.g., combined with deep learning) can also be applied to needs learning due to their similar origins.

However, since a state space equal to the number of demand levels was used to preserve the state, it was equivalent to adding a dimension to the state of reinforcement learning from the original state, making the problem become relatively more complex with the need to occupy more space. This further worsens the space occupation problem of reinforcement learning. In the next step of our research, we intend to follow the example of deep reinforcement learning, trying to use a combination of deep learning methods, such as using neural networks instead of state spaces, to achieve better results. In the aspect of application, we intend to apply needs learning to the field of unmanned driving, hoping that it will yield a better result in terms of the trade-off between driving experience and driving safety.

## 5. Conclusions

Based on Maslow’s hierarchy of needs theory, this paper associated an individual’s movement level with Maslow’s expressive and coping behaviors, proposed a needs learning algorithm that can combine the environment and the agent’s own need to make decisions according to different states, and provided its overall formulation and process. In addition, we designed an experiment to compare needs learning with reinforcement learning. The experimental results showed that needs learning performed better than reinforcement learning in problems that required a trade-off between the agent’s own needs and the target’s needs. Finally, we deployed it on a quadruped robot to generate stable gait. The results showed that needs learning performs well in this task and is also better than reinforcement learning.

To summarize, the needs learning framework pays more attention to the influence of intrinsic motivation on behavior than reinforcement learning. Through a wave model of motivation inspired by the hierarchy of needs theory, we quantified the needs and set a learning mechanism in which the agent’s action corresponded to movement. We also determined that it can be better to simulate multiple levels of needs in human mechanisms to accomplish some tasks. This also provides a relevant basis for further research.

## Figures and Tables

**Figure 1 sensors-22-07302-f001:**
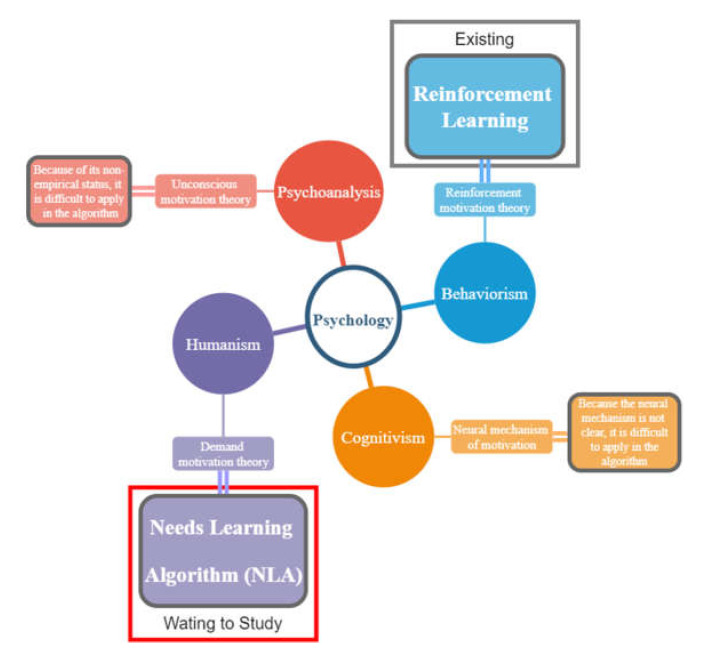
Motivational theoretical basis for the needs learning algorithm.

**Figure 2 sensors-22-07302-f002:**
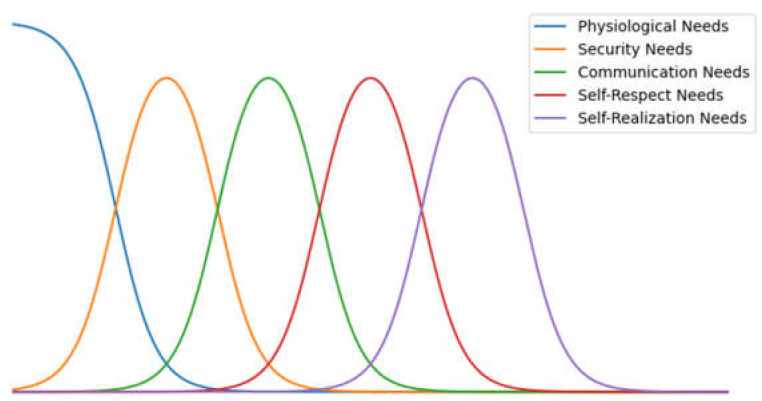
Needs level development relationship.

**Figure 3 sensors-22-07302-f003:**
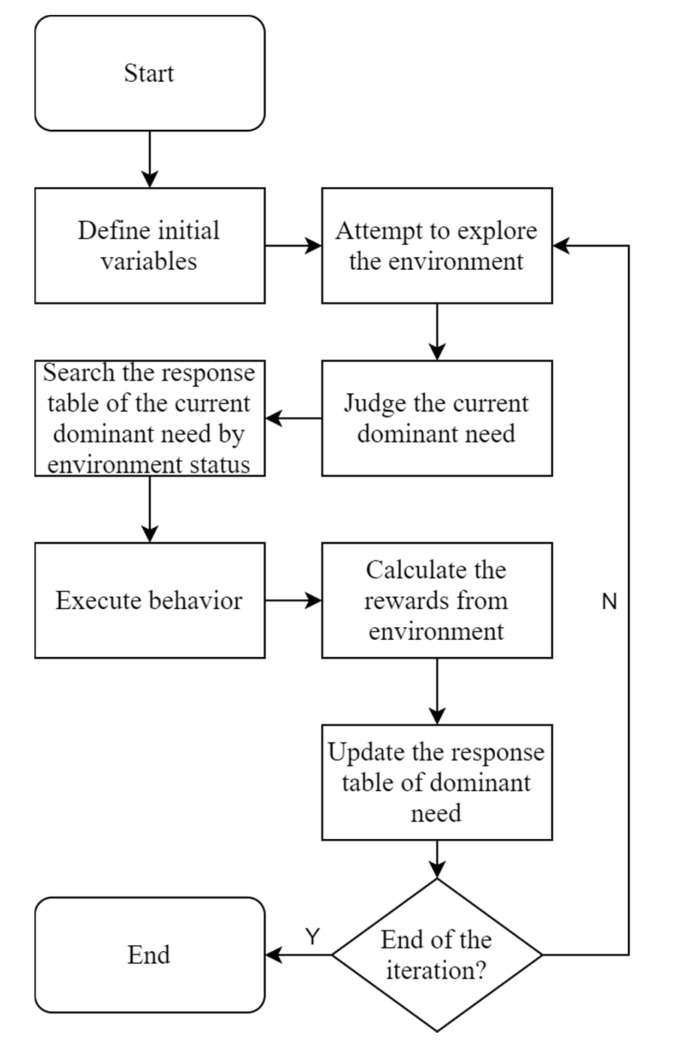
Diagram of algorithm flow.

**Figure 4 sensors-22-07302-f004:**
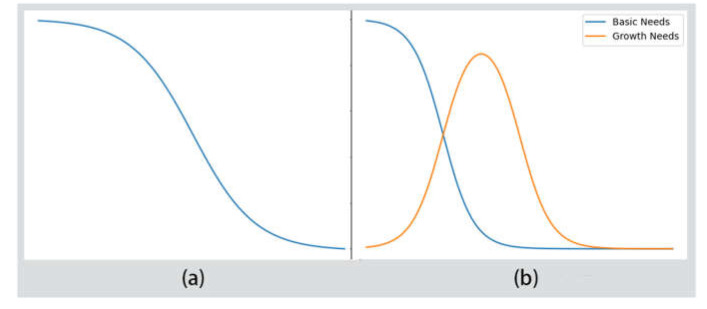
Functional representation of hierarchy of needs. (**a**) The raw wave function; (**b**) graph of the inhibition relationship for different need levels.

**Figure 5 sensors-22-07302-f005:**
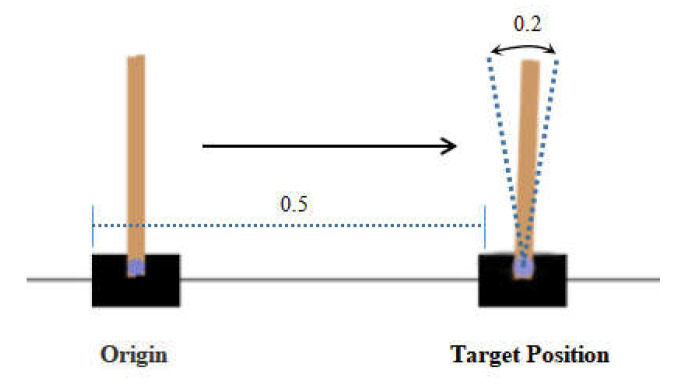
Description of cart pole comparison task.

**Figure 6 sensors-22-07302-f006:**
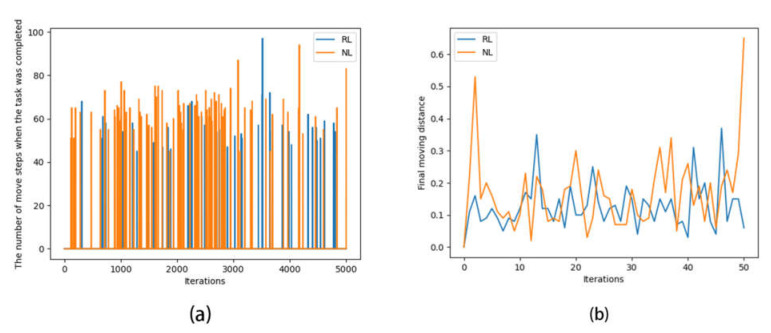
Algorithm comparison results. (**a**) Diagram of the number of moving steps when the task was completed; (**b**) change chart of final moving distance.

**Figure 7 sensors-22-07302-f007:**
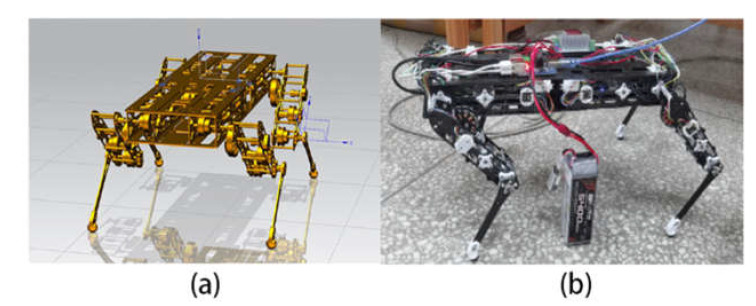
Quadruped robot for experiment. (**a**) Simulation of quadruped robot; (**b**) real robot of quadruped robot.

**Figure 8 sensors-22-07302-f008:**
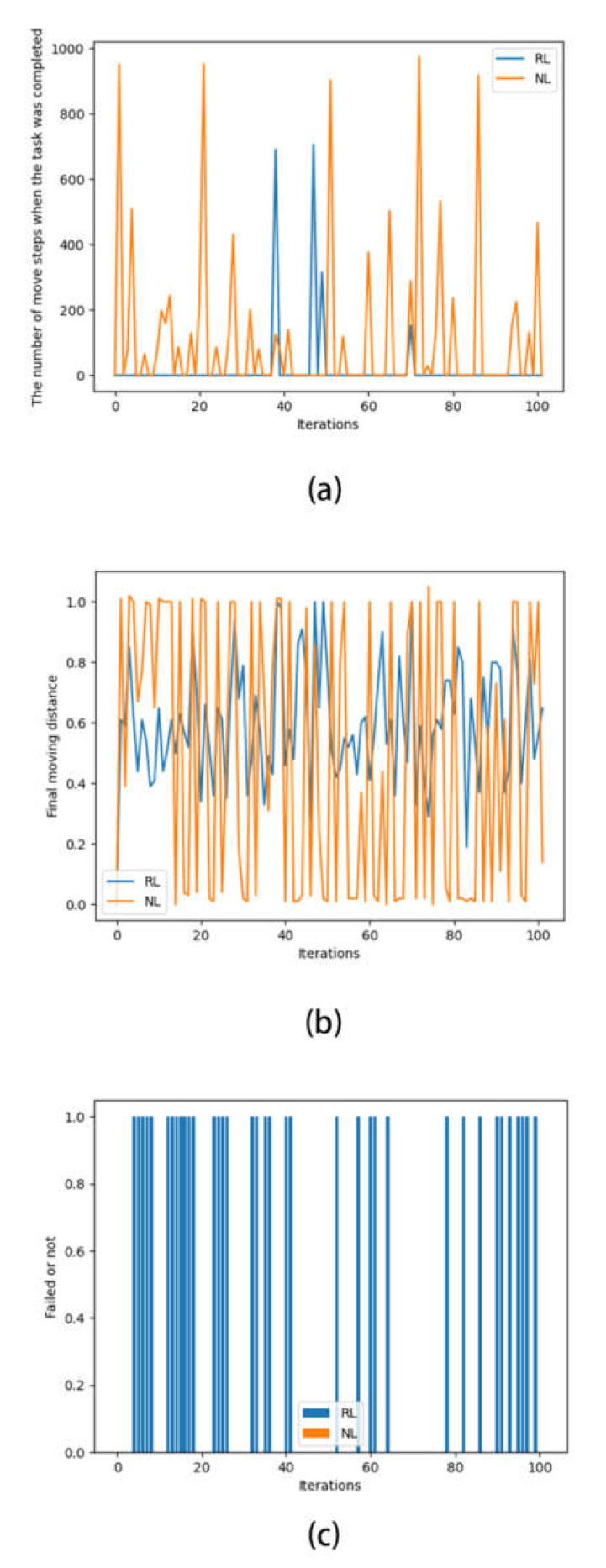
Simulation test results. (**a**) Diagram of the number of moving steps when the quadruped robot movement; (**b**) change chart of final moving distance in comparative test of quadruped robot movement; (**c**) task failure diagram of comparative test of quadruped robot movement.

**Figure 9 sensors-22-07302-f009:**
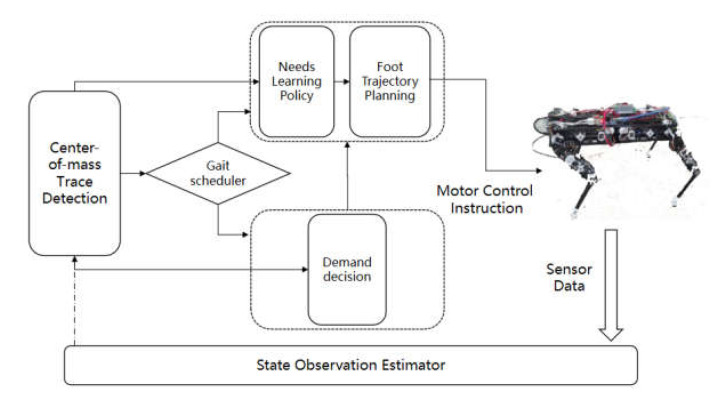
Control block diagram of the quadruped robot control policy.

**Figure 10 sensors-22-07302-f010:**
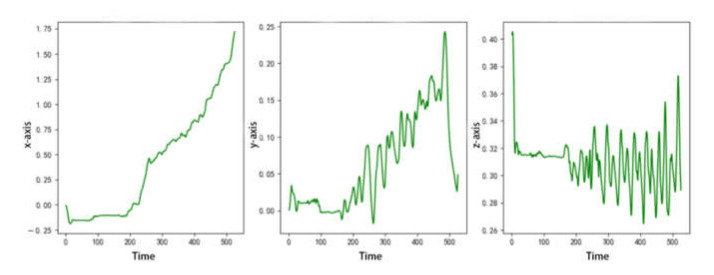
Motion trajectory of quadruped robot using needs learning algorithm.

**Table 1 sensors-22-07302-t001:** Definition of the needs learning vocabulary.

Term	Explanation	Example
Behavior	State transition.	
Motivation	Direction of state transition.	
Objective	Self-realization goals, the results we expected the problem to achieve.	Quadruped robot moves to the end.
Basic Needs	The basic requirement of maintaining a state transition. Its value will inhibit the value of growth needs.	Quadruped robot does not fall.
Growth Needs	Degree of problem solving.	Quadruped robot moves a certain distance.
Dominant Need	A greater value need. It plays a decisive role in motivation.	
Peak Experience	A state of the agent. It is directly proportional to the value of growth needs and will weaken the inhibition of basic needs on growth needs.	
Self-Realization	The highest value of growth needs, the global optimal solution to the problem.	
Expressive Behavior	The most basic behavior of the agent.	Motor rotation of quadruped robot.
Coping Behavior	A complex behavior composed of basic expressive behaviors.	Gait of quadruped robot.

## Data Availability

The data presented in this study are available on request from corresponding author.

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
