# Peer review of "A Needs Learning Algorithm Applied to Stable Gait Generation of Quadruped Robot"

_sensors, 2022, doi:10.3390/s22197302_

Round 1
Reviewer 1 Report
Methods of paper is well written and understanding but application of quadruped control is not understanding.
Quadruped model must be explained.
Quadruped control strategy must me explained by adding a control block diagram.
Control results are not understandable. What is references for control study and Need to see compared results with reference inputs.
If you only want to publish your algorithms effectiveness, you need to change title and journal selection. Ä°f you say control in title readers need yo see detailed control study.
Author Response
Dear Reviewer,
Thank you for your response and comments concerning our manuscript entitled, “A Needs Learning Algorithm Applied to Stable Gait Control of Quadruped Robot”. We have examined the comments carefully and have made corrections that we hope meet with your approval. The revised portions are marked in red in the paper. The main corrections in the paper and the responses to the reviewer’s comments are also described below.
Thank you again.
RESPONSES TO THE REVIEWERS’ COMMENTS
Quadruped model must be explained.
We have added an explanation of this (section 4.2, page 13, line 37 to page 14, line 3).
Quadruped control strategy must me explained by adding a control block diagram.
We have added a control block diagram in (section 4.2, page 14, line 4 to page 14, line 8).
Control results are not understandable. What is references for control study and Need to see compared results with reference inputs.
The reference for comparison is Q-learning, a machine learning method that was first used in the exploration of agent and is commonly used in some studies of quadruped robot control. Some references to its application to quadrupedal robot control are added in section 1 (section 1, page 1, line 39 to page 2, line 1), and the method is reproduced in section 4.1 to compare it with needs learning under the same conditions (section 4.1, page 11, line 35 to page 12, line 2).
If you only want to publish your algorithms effectiveness, you need to change title and journal selection. Ä°f you say control in title readers need yo see detailed control study.
A control block diagram has been added to explain the quadruped control policy (section 4.2, page 14, line 4 to page 14, line 8). Also, considering that our method is indeed different from "control", we have changed the title from "stable gait control" to "stable gait generation"(Title).
Reviewer 2 Report
The work entitled “A Needs Learning Algorithm Applied to Stable Gait Control of Quadruped Robot” present a need-based algorithm for ML for gait stabilization on a quadruped robot.
The work is well structured however, there are a few points that should be addressed:
_ the format of the manuscript does not seem to be generated using the journal templates.
_Figure 2, 4 does not report any unit or label for the X-Y axis.
_Overall captions should be developed further as well as add reference to sub-parts of the figure e.g. a, b in figure 4, 6, 7 a,b,c fig 8
_ format of equations / mathematical references is not aligned with the normal text.
_ relevant equations should be numbered and referenced
_figure 6 can be enlarged and take-home message in the caption made clearer.
_figure 9 a motion trajectory is unclear. A better display option should he chosen. E.g. XY (is z relevant? If so, can be on a separate graph or all combined in a 3d graph)
_overall, a supporting video of the experiment would have improved clarity of the work.
_ references can be improved.
_ conclusion and discussion should summarize the most important results providing quantitative support to the claims.
_ in several plots the label refers to RL and NL, those are not properly introduced in the text?
_ the claims in the discussion/conclusion that about that the needs learning performed better than reinforcement. However, from the results presented is very difficult to appreciate. For example, figure 8 c does not show any NL as well as is not clear with the label failed or not if the plot shows success or fail. The small description (which should be improved) make me incline to think to the latter so NL never failed? Can be an experimental motion trajectory be presented using both and not presenting only the one with NL.
_Furthermore, the many iterations are not completely presented, variation, statistical and experimental analysis should be presented.
Author Response
Dear Reviewer,
Thank you for your response and comments concerning our manuscript entitled, “A Needs Learning Algorithm Applied to Stable Gait Control of Quadruped Robot”. We have examined the comments carefully and have made corrections that we hope meet with your approval. The revised portions are marked in red in the paper. The main corrections in the paper and the responses to the reviewer’s comments are also described below.
Thank you again.
RESPONSES TO THE REVIEWERS’ COMMENTS
the format of the manuscript does not seem to be generated using journal templates.
We have completed the changes using the journal template.
Figure 2, 4 does not report any unit or label for the X-Y axis.
Figures 2 and 4 are redrawn from the original image of Maslow's wave model of the hierarchy of needs theory. He did not give a specific x and y axis when he presented such an image, but only used such a trend to explain the content of the wave model figuratively.
Overall captions should be developed further as well as add reference to sub-parts of the figure e.g. a, b in figure 4, 6, 7 a,b,c fig 8
The sub-parts of each figure have been cited in our paper with reference to other Sensors accepted papers (fig. 4, 6, 7, 8).
format of equations / mathematical references is not aligned with the normal text.
We have revised this in our paper through the formula editor.
relevant equations should be numbered and referenced
We have revised this in our paper (section 2.2, page 5; section 2.2.1, page 7; section 3, page 9)
figure 6 can be enlarged and take-home message in the caption made clearer.
We have revised this in our paper (fig. 6).
figure 9 a motion trajectory is unclear. A better display option should he chosen. E.g. XY (is z relevant? If so, can be on a separate graph or all combined in a 3d graph)
The z-axis data is relevant and is used to show the smoothness of the quadruped robot while moving. We make a description (section 4.2, page 14, line 15 to line 18). However, after adding the z-axis data, the dimension of the motion description becomes four-dimensional (x, y, z, t), which is difficult to visualize using a single image. A more intuitive description of the motion trajectory will be shown in the attached video.
overall, a supporting video of the experiment would have improved clarity of the work.
The attachment upload in Sensors requires all reviewers' comments to be responded to, so I cannot provide a link to the supporting video immediately; you will see the link to the supporting video in the Supplementary Materials section of the new version of the manuscript later.
references can be improved.
We have revised this in our paper (section 1, page 1, line 39 to page 2, line 1; section 4.2, page 13, line 37 to line 40).
conclusion and discussion should summarize the most important results providing quantitative support to the claims.
Furthermore, the many iterations are not completely presented, variation, statistical and experimental analysis should be presented.
In sections 4 and 5, we have added some new descriptions to summarize and discuss these results (section 4, page 15, line 8 to line 13; section 5, page 15 line 30 to line 34).
in several plots the label refers to RL and NL, those are not properly introduced in the text?
RL refers to reinforcement learning, NL refers to needs learning. We have added a specific description before using these abbreviations for the first time (section 3, page 10, line 3 to line 4).
the claims in the discussion/conclusion that about that the needs learning performed better than reinforcement. However, from the results presented is very difficult to appreciate. For example, figure 8 c does not show any NL as well as is not clear with the label failed or not if the plot shows success or fail. The small description (which should be improved) make me incline to think to the latter so NL never failed? Can be an experimental motion trajectory be presented using both and not presenting only the one with NL.
In Figure 8(c), NL is not shown because it did not happen to let itself fall during a finite number of actions in each iteration. Therefore, we believe that NL is more effective in weighing the need for survival (not falling) and task (moving forward) compared to RL. However, we did have an unclear formulation, for which we modified the description (section 4.1, page 13, line 21 to line 27).
Also in the experiments the motion trajectories generated by the two methods are indirectly represented in Figure 8(b), which is the final distance moved by the agent at the completion of each iteration. Here too, our formulation is unclear, for which we modified the description (section 4.1, page 13, line 12 to line 20).
Round 2
Reviewer 1 Report
Thanks for revision.
Reviewer 2 Report
Overall the issues highlighted have been addressed.